# Mid-infrared single-pixel imaging at the single-photon level

Yinqi Wang[1], Kun Huang [1,2,3] ✉, Jianan Fang[1], Ming Yan[1,2], E Wu[1,2] & Heping Zeng [1,2,4,5] ✉

Single-pixel cameras have recently emerged as promising alternatives to multi-pixel sensors due to reduced costs and superior durability, which are particularly attractive for mid-infrared (MIR) imaging pertinent to applications including industry inspection and biomedical diagnosis. To date, MIR single-pixel photon-sparse imaging has yet been realized, which urgently calls for high-sensitivity optical detectors and high-fidelity spatial modulators. Here, we demonstrate a MIR single-photon computational imaging with a single-element silicon detector. The underlying methodology relies on nonlinear structured detection, where encoded time-varying pump patterns are optically imprinted onto a MIR object image through sum-frequency generation. Simultaneously, the MIR radiation is spectrally translated into the visible region, thus permitting infrared single-photon upconversion detection. Then, the use of advanced algorithms of compressed sensing and deep learning allows us to reconstruct MIR images under sub-Nyquist sampling and photon-starving illumination. The presented paradigm of single-pixel upconversion imaging is featured with single-pixel simplicity, single-photon sensitivity, and room-temperature operation, which would establish a new path for sensitive imaging at longer infrared wavelengths or terahertz frequencies, where high-sensitivity photon counters and high-fidelity spatial modulators are typically hard to access.

Mid-infrared (MIR) imaging becomes a key enabler of great scientific and technical interest in a variety of applications, such as biomedical diagnosis, defect inspection, molecular spectroscopy, and remote sensing[1,2]. In these envisioned scenarios, sensitive MIR response is highly demanded to access dramatically improved performances in terms of detection sensitivity, working distance, or noninvasive capability[3–5], which is particularly pertinent to low-photon-flux contexts, for instance, trace characterization of photosensitive materials, penetration imaging through scattering media, and phototoxicity-free examination for biological samples. However, the imperious call for highly sensitive MIR imagers challenges conventional focal plane arrays (FPAs) that still face several technical limitations including high dark noise, low pixel count, and thermal susceptibility[6,7]. Furthermore, the FPAs are typically subjected to expensive fabrication processes, cryogenic working conditions, and stringent end-user controls. Notably, emerging low-dimensional materials, like colloidal quantum dots[8], black phosphorus[9], graphene[10], and tellurium nanosheet[11], hold great promise to sense infrared photons at room temperature, albeit with pressing difficulties in dark-current suppression to improve the sensitivity and large-area deposition to increase the pixels[12,13]. To date, it remains a long-standing quest to achieve single-photon MIR direct imaging at room temperature.

[1]State Key Laboratory of Precision Spectroscopy, East China Normal University, 200062 Shanghai, China. [2]Chongqing Key Laboratory of Precision Optics, Chongqing Institute of East China Normal University, 401121 Chongqing, China. [3]Collaborative Innovation Center of Extreme Optics, Shanxi University, 030006 Taiyuan, Shanxi, China. [4]Jinan Institute of Quantum Technology, 250101 Jinan, Shandong, China. [5]Shanghai Research Center for Quantum Sciences, 201315 Shanghai, China. ✉e-mail: khuang@lps.ecnu.edu.cn; hpzeng@phy.ecnu.edu.cn

In recent years, the so-called single-pixel cameras provide an alternative imaging architecture by using single-element detectors in conjunction with spatial encoding masks[14–17]. Specifically, a spatial light modulator (SLM) is placed before or after the targeted scene to generate a sequence of well-defined patterns, and the correlated intensity is synchronously measured by a detector without spatial resolution[18]. Such a computation imaging modality offers practical and economic advantages by excluding the need for slow mechanical scanning or expensive multipixel detectors[19,20]. Another distinct feature for single-pixel detector is the much faster time response comparing to the pixelated imaging device, thus favoring the time-resolved imaging or high-resolution surface profiling[21–23]. Moreover, the single-pixel imaging approach can be compatibly enhanced by adopting sophisticated algorithms of compressed sensing[24] and machine learning[25], which enables us to realize high-frame-rate videography with sub-Nyquist sampling[26] and low-light-level visualization with limited photons[27]. Nevertheless, the single-element scheme for photon-sparse imaging is so far restrictedly operated at the visible or near-infrared spectral bands due to the accessibility of high-performance optical detectors and spatial modulators[16]. Nowadays, there is a significant impulse to extend the operation wavelength into the MIR region regarding the aforementioned applications.

Pioneering demonstrations of MIR compressive imaging are reported by using few-pixel infrared sensors[28–30], yet with a sensitivity far way from the single-photon level. In addition to the lack of single-photon detectors, the other constraint to approach single-pixel imaging at the MIR region lies in the operation wavelength range of conventional SLMs based on liquid crystal or electromechanical micromirror[31]. Although the reflectivity of the micromirrors extends far into the infrared, the ability of the digital micromirror devices (DMDs) to apply an intensity modulation is limited by the parasitic diffraction effects that dominate over the beam steering especially at longer wavelengths[16]. In parallel, recent technological advances in graphene metasurfaces allow to realize high-speed MIR modulators[32], but still in the infancy stage with a prototype form of only 6 × 6 functional pixels[33]. So far, MIR single-photon computational imaging has yet been realized to reveal the full potential of the single-pixel paradigm, which thus urgently calls for the development of techniques to address the challenges on the single-photon detection and high-resolution modulation at MIR wavelengths.

Here, we devise and implement a frequency-upconversion approach for single-pixel MIR imaging at the single-photon level. In our methodology, the object scene is spatially interrogated by a near-infrared structured pump within a nonlinear crystal to perform the sum-frequency generation. The spatial manipulation on the pump is realized using comparatively economic and well-developed optical SLMs. The resulting optical modulation provides an indirect way to access high-resolution dynamic patterns onto the MIR radiation. Simultaneously, the involved nonlinear conversion facilitates spectral translation of screened MIR field into the visible region, where a high-performance silicon detector can be used for efficient and sensitive registration of upconverted photons. Consequently, knowledge of the projected patterns and of the corresponding measured intensities are combined to reconstruct the object image. Furthermore, a real-time video is recorded at a frame rate of 10 Hz for a pixel number of 16 × 16. Notably, the imaging sensitivity is significantly improved by the coincidence pulsed pumping with a spectral-temporal optimization, which enables us to demonstrate the single-photon imaging with an illumination intensity down to 0.5 photons/pulse. In addition, MIR compressive imaging at the single-photon level is realized with an undersampling ratio of 25% by leveraging a numerical denoiser via deep convolutional neural network. The presented single-pixel imaging paradigm is featured with upconversion structured detection at the single-photon level, which would open up new possibilities in MIR applications at low-photon-flux scenarios, for instance in covert imaging and biological imaging.

## Results
### Basic principles

The core of the MIR single-pixel imaging is to devise a proper method to prepare a sequence of dynamic and deterministic patterns. Typically, the SLM devices based on liquid crystals or micromirror arrays are optimized to operate at the visible or near-infrared regime[17]. Indeed, the beam steering performance of the modulator is fundamentally limited by the parasitic diffraction effects at longer infrared wavelengths, which results in a reduced spatial modulation accuracy[16]. An alternative promising approach to prepare high-resolution patterns may resort to the optically controlled modulation[34], where the electromagnetic radiation is spatially manipulated within an intermediate medium excited by a shorter-wavelength pumping. This methodology has been used to implement terahertz single-pixel imaging with a sub-wavelength resolution[35]. Indeed, the spatial resolution of the imaging is determined by the optical pump, thus overcoming the constraint of the diffraction limit. Here, we propose to use a nonlinear frequency-upconversion process to facilitate the MIR optical spatial modulation. The simultaneous consequence of spectral transduction renders this scheme different from previous reports on the optically controlled modulators[36–38], which favors subsequent single-photon detection with a silicon bucket detector.

Specifically, the proposed modality for nonlinear structured detection is based on the sum-frequency generation (SFG). The SFG field $E_{\mathrm{up}}$ can be deduced from three-wave coupling equations under approximations of paraxial interaction and slowly varying envelope[39,40]:

$$\frac{\partial E_{\mathrm{up}}(x,y)}{\partial z} = \frac{i2\omega_{\mathrm{up}}^2 d_{\mathrm{eff}}}{k_{\mathrm{up}}c^2} \times E_s(x,y)E_p(x,y)e^{-i\Delta k_z z}, \qquad (1)$$

where $E_s$ and $E_p$ are the signal and pump electric fields, $d_{\mathrm{eff}}$ denotes the effective nonlinear coefficient, $c$ is the speed of light in vacuum, $z$ is the propagation direction, and $\Delta k_z$ represents the longitudinal phase mismatch of the nonlinear conversion process. The involved angular frequencies for the signal, pump and SFG fields satisfy the energy conservation as $\omega_{\mathrm{up}} = \omega_s + \omega_p$. In practice, the conversion efficiency is usually optimized at the phase-matching condition with $\Delta k_z = 0$, the SFG intensity is thus simply related to the inner product of the signal and pump intensities, as given by $I_{\mathrm{up}}(x, y) \propto I_s(x, y) \times I_p(x, y)$[41,42]. Such a relation indicates that the pump intensity pattern serves as a spatial transmission mask onto the MIR object image, and the mask filtered pattern is directly linked to the intensity profile of the upconverted field. With this approach, any optical pattern generated through a well-developed SLM device can act as a pumping source to shape the incident MIR radiation. Moreover, the power of SFG light $P_{\mathrm{up}} = \iint I_{\mathrm{up}}(x, y)\mathrm{d}x\mathrm{d}y$ detected by a single-element detector is correlated to each optical pump pattern. Given a sequence of orthonormal patterns $I_p^{(\mathrm{m})}(x,y)$, where $m$ is the pattern sequence number, the corresponding differential intensity signals between the positive and inverse patterns $P_{\mathrm{up}}^{(\mathrm{m})}$ are measured to provide weighted coefficients in the subsequent image estimation. Based on $M$ patterns, the two-dimensional image of the object $O(x, y)$ can be reconstructed by

$$O(x,y) = \frac{1}{M} \sum_{m=1}^{M} P_{\mathrm{up}}^{(\mathrm{m})} I_p^{(\mathrm{m})}(x,y). \qquad (2)$$

Consequently, the nonlinear spatial modulation provides all-optical solution to facilitate the upconversion structured detection, which lays the foundation to implement highly sensitive single-pixel MIR imaging

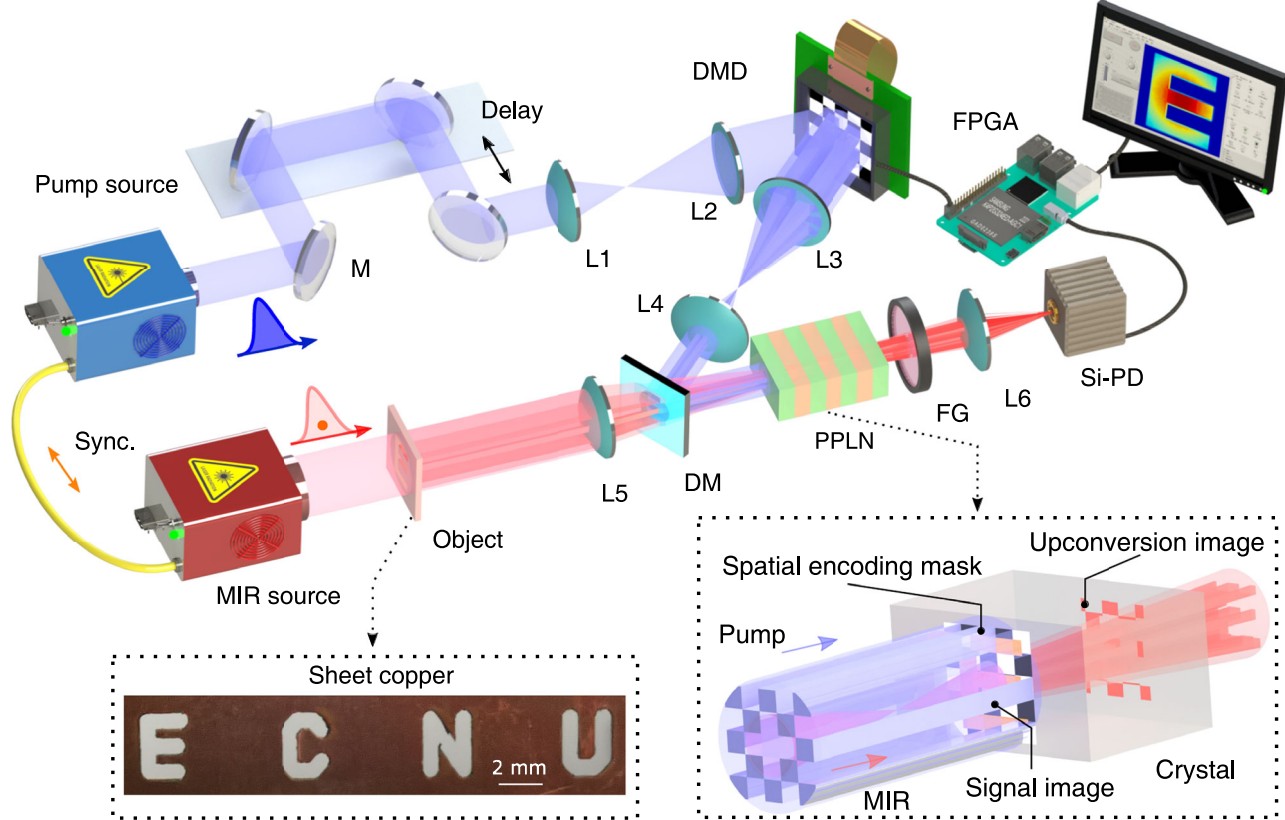

**Fig. 1 | MIR single-pixel imaging based on nonlinear structured detection.** A collimated MIR pulse at 3070 nm is launched at a transmission mask made of a piece of sheet copper carved with four letters. The resulting object image is projected by a lens into the center of a periodically poled lithium niobate (PPLN) crystal. Simultaneously, the nonlinear crystal is pumped by a synchronous structured field at 1030 nm. The pump beam is spatially patterned by a digital micro-mirror device (DMD), and then combined by a dichroic mirror (DM) with the MIR light to perform the sum-frequency generation. Through the nonlinear interaction, a sequence of well-defined pump patterns are optically imparted into the MIR radiation. Meanwhile, the nonlinear spatial mapping also results in a frequency upconversion of MIR photons into the visible band. Consequently, the upconverted photons are collected by a single-pixel silicon-based photodiode (Si-PD) after a spectral filtering group (FG). Finally, the intensity measurements correlated with the time-varying patterns enables us to reconstruct the MIR object image. All the electronic timing and data acquisition is assisted by a controlling unit based on a field programmable gate array (FPGA). A conceptual zoom-in for the structured upconversion within the nonlinear crystal is shown in the bottom-right inset.

with a silicon photon-counting detector. More discussion on the theoretical model is given in Supplementary Note 1.

## Experimental setup

Figure 1 presents the artistic illustration of the experimental setup for the MIR single-pixel imaging based on nonlinear structured detection. The MIR signal and pump pulses are prepared from a synchronized dual-color ultrafast fiber laser system (see the "Method" section and Supplementary Note 2). The MIR beam at 3070 nm is enlarged by a beam expander before illuminating a transmission object mask engraved with four letters. The formed object image is transferred by a lens into a periodically poled lithium niobate (PPLN) crystal to perform the nonlinear upconversion imaging. The object image is scaled down to be accommodated with the geometric dimensions of the PPLN channel of $10 \times 1 \times 1$ mm$^3$ (length × width × thickness). The synchronized pump pulse at 1030 nm is spatially combined by dichroic mirror into the PPLN crystal to facilitate the nonlinear spatial modulation. Figure 2a–c shows three representative Hadamard patterns loaded onto the DMD, which correspond to pump intensity distributions at the center of the nonlinear crystal as shown in Fig. 2d–f. The prepared optical patterns constitute high-resolution masks for the MIR radiation through the nonlinear three-wave mixing. The nonlinear spatial mapping is further verified by the measured patterns for the upconverted SFG beam as presented in Fig. 2g–i. Note that the presented optical patterns are corrected by the inhomogeneous beam profile that is

measured at the all-on state of the DMD (see Supplementary Note 4). Moreover, the optical masks for a full set of Hadamard matrices are recorded for comparison, as given in Supplementary Movie 1.

To facilitate the optical spatial modulation, the beam size of the structured pump is adapted to cover the object image, meanwhile the beam diameter should be confined within the cross section of the nonlinear crystal. The operation temperature of the PPLN crystal is stabilized at 48.6 °C for a poling period of 20.9 $\mu$m to approach the phase-matching condition. Furthermore, the conversion efficiency is optimized by carefully tuning the temporal delay between the MIR and pump pulses. The employed coincidence pumping scheme leverages the high peak power and ultrashort excitation window of ultrashort pulses, which helps to increase the conversion efficiency and reduce the background noise[43–46]. Then, the generated SFG signal at 771 nm is steered through a spectral filtering stage to remove the background noise from pump-induced parametric fluorescences[47,48]. The filtered photons are finally collected by a free-space silicon-based photodiode. The involved timing control and data acquisition are implemented by a high-precision digital module based on a field programmable gate array (FPGA). The operating configuration and timing sequence are detailed in the "Method" section and Supplementary Note 3.

## Single-pixel MIR imaging

Now we turn to characterize the single-pixel imaging performance by starting with Hadamard patterns. In comparison to the raster scanning,

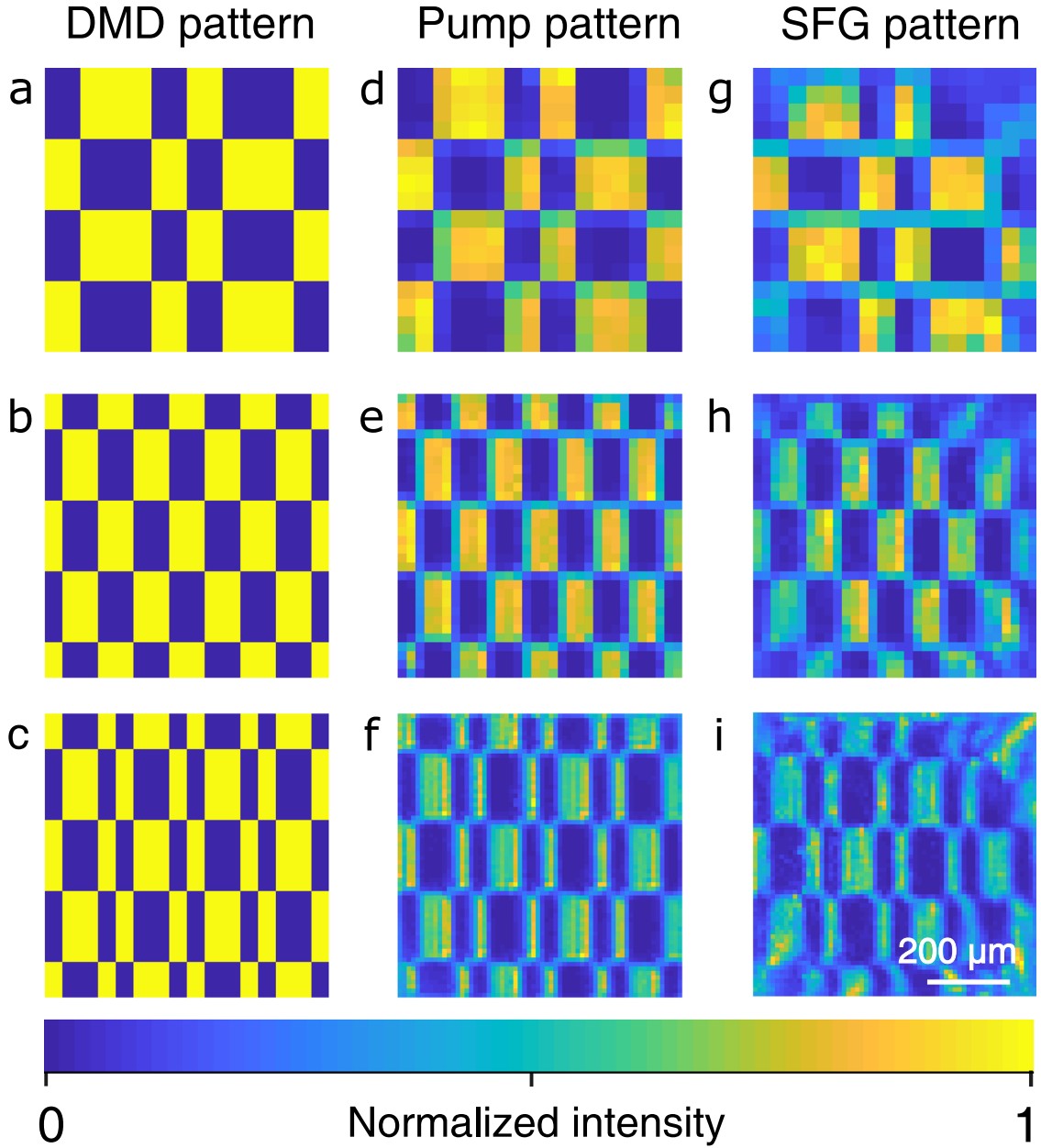

**Fig. 2 | Nonlinear structured modulation of optical pump patterns.**
**a–c** Exemplary Hadamard patterns written onto the spatial light modulator.
**d–f** Corresponding intensity distributions of the pump at the nonlinear crystal.

**g–i** Spectrally upconverted patterns through the sum-frequency generation. The correspondence for the full set of Hadamard patterns is verified in Supplementary Movie 1.

multipixel masking schemes offer the advantage of minimizing the effect of the detector noise by using more light in each measurement[16], which helps to improve the imaging sensitivity (see Supplementary Note 5). The Hadamard matrix is a square matrix whose entries are either +1 or −1 and whose rows are mutually orthogonal. Consequently, the Hadamard encoding is featured to provide an orthonormal set that helps to reduce the mean squared error in each pixel[35]. This unique property of orthonormality renders the Hadamard matrix useful in various sensing and imaging applications[17]. Moreover, the Hadamard sampling enables us to perform image reconstruction by a computationally fast algorithm with a simple matrix transpose operation (see Method Section).

Figure 3a–d presents the numerical simulations for the single-pixel upconversion imaging. Correspondingly, the reconstructed object images based on experimental data are presented in Fig. 3e–h, i–l for resolvable pixels of 16 × 16 and 64 × 64, respectively. Here, the

weighted signal is obtained by measuring the differential intensity for each pattern and its contrast inverse, which enables us to remove any offset in the image due to background noise and suppress low-frequency oscillations for the illumination source[16]. The exhibiting blurring effect is due to the reduced numerical aperture determined by the phase-matching bandwidth of the angular acceptance for the nonlinear crystal. Additionally, the limited spatial frequency bandwidth of the upconversion imaging system inevitably results in an imperfect mapping of the optical pump masks, which may contribute to the image degradation in the single-pixel imaging paradigm. In our experiment, the pixel size for the prepared optical mask is 10.8 $\mu$m, which is much smaller than the achieved resolution about 91 $\mu$m. An improved space-bandwidth product can be achieved by optimizing the upconversion imaging system[48]. In the experiment, the upconverted SFG intensity is measured by a high-speed silicon photodetector with a minimum noise equivalent power of about 3 pW/Hz$^{1/2}$. The frequency-

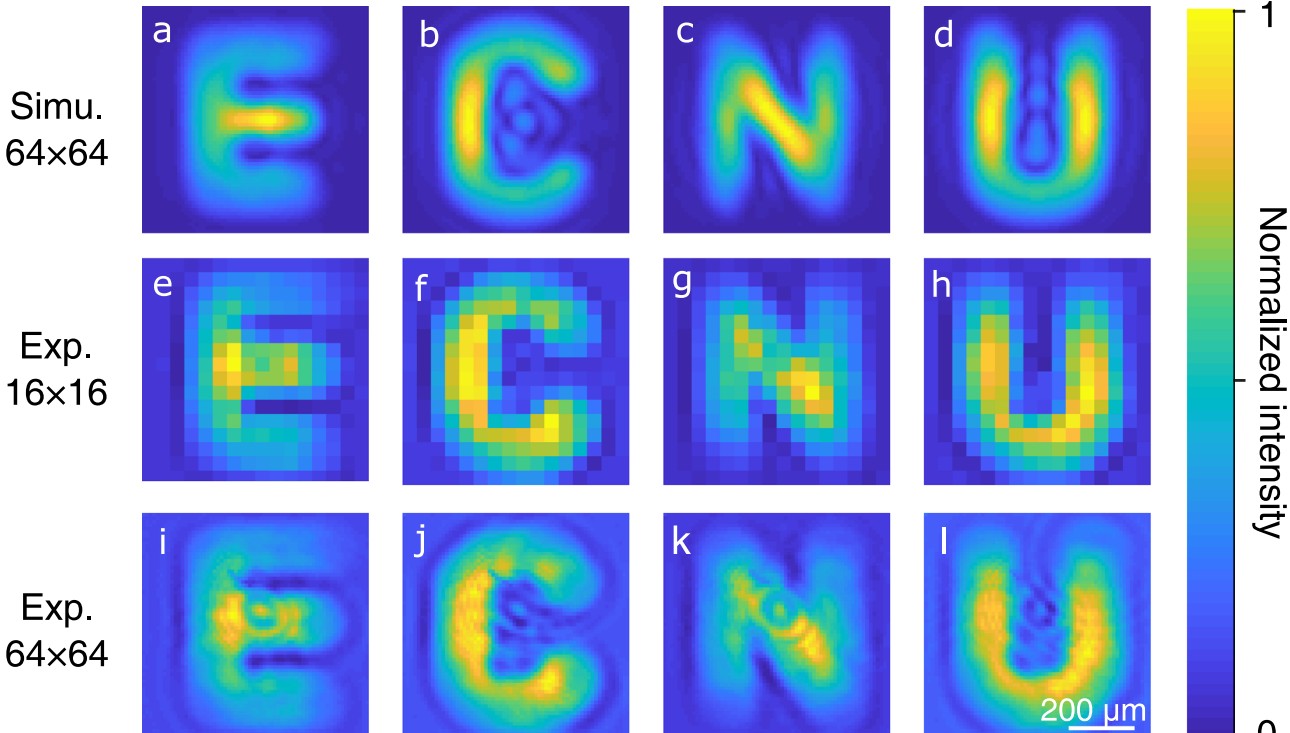

**Fig. 3 | MIR single-pixel imaging based on Hadamard encoding. a–d** Numerical simulations on the single-pixel upconversion imaging for the transmission object with four letters as shown at the bottom-left corner in Fig. 1. **e–l** Experimentally reconstructed object images with resolvable pixels of 16 × 16 (**e–h**) and 64 × 64 (**i–l**). The use of Hadamard patterns as the decomposition basis favors implementing image reconstruction with a computationally fast algorithm. Consequently, a real-time MIR single-pixel imaging at 10 frames per second is demonstrated for 16 × 16 reconstructed pixels, which is illustrated by the recorded video referred to Supplementary Movie 2.

upconversion strategy is a key to realizing the sensitive and high-speed detection, which enables us to achieve real-time imaging at a frame rate of 10 Hz for 16 × 16 pixels (see Supplementary Movie 2).

## MIR photon-sparse imaging

Next, we investigate the MIR single-pixel imaging in the ultralow photon flux regime. To this end, the bucket detector is replaced by a single-photon-counting module (SPCM) based on a silicon avalanche photodiode. It is the upconversion structured detection that makes it possible to leverage single-photon visible detectors with high efficiency, low noise, and fast speed, thereby providing an effective solution to the sensitive MIR single-pixel imaging. In our experiment, the frequency upconverter is implemented with narrow-band signal illumination, which allows the use of stringent spectral filtering to suppress the pump-induced noise. Moreover, the instantaneous response of the nonlinear upconversion enables us to apply the synchronous ultrashort optical gating for the signal detection, thus significantly reducing the background noise from the ambient random scattering[43,44]. The resultant suppression of background noises in the temporal and spectral domains helps to realize MIR single-photon computation imaging as shown in Fig. 4. Particularly, the MIR signal pulse is attenuated to 100 photons/pulse before being injected into the imaging system, and the integration time is set to 5 ms for each pattern. The reconstructed images are presented in Fig. 4a, f for pixel numbers of 16 × 16 and 64 × 64, respectively. As shown in Fig. 4e, j, further reduction of the photon number to 0.5 is still permitted to obtain high-contrast images albeit with a longer accumulation time up to 1 s. For a fair comparison, here we acquire the same amount of photons for each pattern. It can be seen that the increase in the number of pixels leads to increased image noise due to the less photons distributed for each individual pixel. Additionally, more dark noises are included for a longer exposure time, thus reducing the

signal-to-noise ratio of the intensity measurement. The degradation of image quality is especially noticeable at the settings of more reconstructed pixels and fewer incident photons. In the single-photon regime, photon-counting fluctuations become prominent due to the intrinsic Poissonian photon statistics for the coherent-state light[23]. The photon-counting stability can be improved by a factor proportional to the square root of the photon number within the integration time. Precise intensity measurement with stable photon-counting rates plays an important role in optimizing the single-pixel imaging quality as investigated in Supplementary Note 6.

As a proof-of-principle demonstration, we apply the implemented sensitive single-pixel camera to examine the images engraved on a silicon substrate in the presence of single-photon illumination power (see Supplementary Note 7). Pertinent to the transparency window for semiconductor materials, the sensitive MIR imaging would be useful in noninvasive chip defect inspection.

## Single-photon compressive imaging

In the above approach, the number of measurements is required to be identical to the pixel number of the reconstructed image. However, more sophisticated strategies based on compressed sensing algorithms enable us to stably reconstruct an image of the scene from fewer measurements than the number of reconstructed pixels, that is in the manner of sub-Nyquist acquisition[24]. In this case, random masks play a key role for the compressed sensing, making each measurement a random sum of pixel values taken across the entire image. Indeed, random patterns constitute a basis that is incoherent with the spatial properties of the image, such that each measurement provides a little information about every pixel.

Here, we use a primal-dual algorithm to recovery the compressed images (see Method Section and Supplementary Note 5 for details). Figure 5a, b shows the compressive imaging performance at the few-

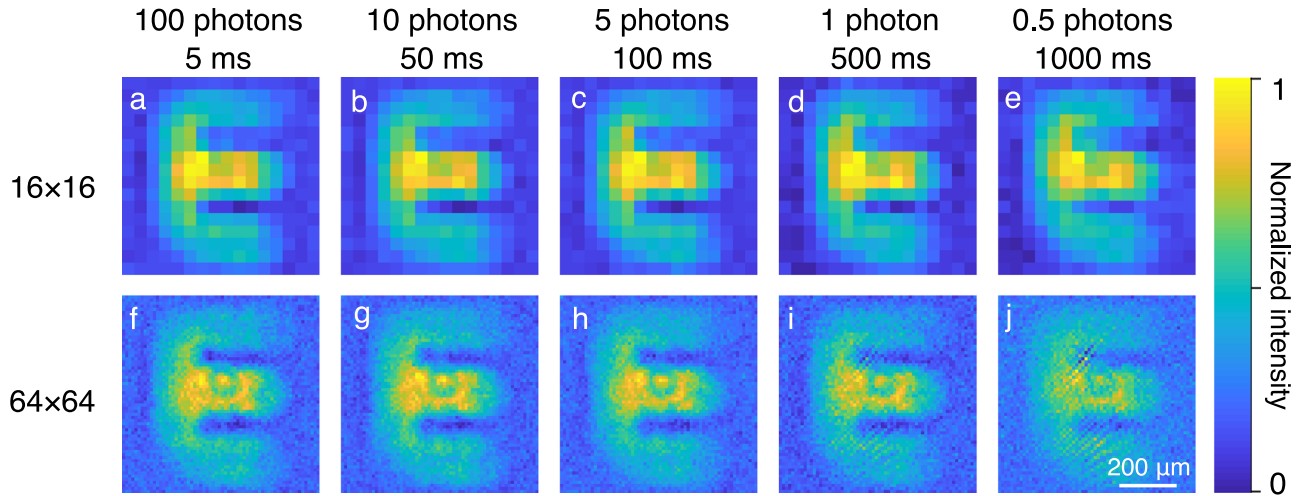

**Fig. 4 | MIR photon-sparse single-pixel imaging.** The MIR single-pixel imaging is performed with a silicon single-photon-counting module in the framework of the nonlinear structured detection. **a–e, f–j** Reconstructed images at various photon fluxes from 100 to 0.5 photons/pulse, under increased accumulation time from 5 ms to 1 s for each masking pattern. The top (**a–e**) and bottom (**f–j**) panels correspond to the number of pixels of 16 × 16 and 64 × 64, respectively.

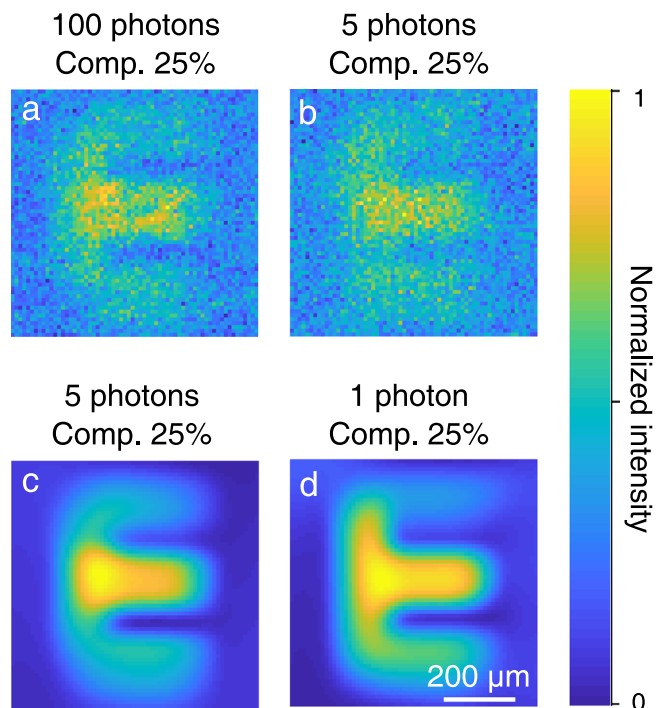

**Fig. 5 | MIR single-photon compressive imaging. a, b** Reconstructed images under the sub-Nyquist sampling with a compression ratio of 25% for MIR illumination intensity of 100 and 5 photons/pulse, respectively. **c, d** Reconstructed images by using a numerical Gaussian denoiser based on deep-learning convolutional neural networks, which permits to obtain high-contrast images for a low-light flux down to 1 photon/pulse. Note that random patterns are used here to perform the compressive sensing algorithm, and each image contains 64 × 64 pixels.

photon level of 100 and 5 photons/pulse, respectively. The letter "E" can be identified even when only 25% of the measurements are used, albeit with the presence of noisy points due to the undersampling. To suppress the image noise, we further adopt a denoiser prior designed for plug-and-play image restoration, which is initially developed in ref. [49] based on deep-learning convolutional neural networks. The restored image is presented in Fig. 5c, which is featured with the almost absence of random noises. Moreover, the noise filtering capability of the effective deep denoiser is manifested in the case of extremely low flux with 1 photon/pulse for an undersampling ratio of 25% as shown in Fig. 5d. Smaller photon numbers and lower undersampling ratio are possible by using more advanced imaging processing algorithms, but at the price of longer image processing time[16,17].

## Discussion

Although the single-pixel architectures are widely demonstrated to perform imaging over a broad electromagnetic spectrum from X-ray to millimeter waves, yet single-pixel cameras with single-photon sensitivity have long been restricted to operate at the visible or near-infrared wavelengths due the availability of highly sensitive optical detectors. Our work addresses the long-sought-after goal to extend the operation wavelength of single-photon computational imaging into the MIR regime. The presented approach of the nonlinear structured detection not only facilitates the high-fidelity mapping of deterministic transmission patterns, but also enables the sensitive upconversion detection for the masked infrared signal with a silicon-based visible photodiode operating at room temperature. The efficient spectral transfer of the spatial information and the ability to count single photons allow the realization of MIR single-pixel imaging under the single-photon-level illumination. Notably, the unique feature of dual functionalities renders our optical pumping configuration conceptually different from the reported optically controlled spatial modulators[34] based on photoexcited substrates[35,36], electrooptic media[37], or nonlinear crystals[38]. In the previous instantiations, no spectral transduction is involved for the optically induced modulation, hence the detection sensitivity of intensity measurement remains limited by the conventional detectors.

The frequency-upconversion detection provides an effective way to realize the MIR single-photon imaging, as manifested in the infrared upconversion imaging systems based on coherent frequency conversion[42,47,48] or quantum spectral correlation[50,51], which typically needs a FPA based on electron multiplying charge-coupled devices (EMCCDs). In comparison, the proposed single-pixel scheme offers advantages of single-pixel simplicity and low cost by using a silicon avalanche photodiode at room temperature. More importantly, the operation bandwidth for single-element detectors and corresponding timing electronics are vastly superior to that for current pixelated devices[16]. The better timing resolution favors the implementation of time-of-flight ranging or three-dimensional

imaging, where the information of precise arrival time for detected photons is required[21].

We note that the achieved spatial resolution is technically limited by the numerical aperture of the upconversion imaging system, where the acceptance angle for the incident light is truncated by the phase-matching angular bandwidth of the nonlinear crystal[39]. Additionally, the field of view is determined by the pump diameter and/or crystal aperture size, whichever is smaller. Hence, the use of a shorter crystal with a wider section would lead to a better spatial resolution and a larger field of view. An alternative approach to enhance the spatial resolution can resort to a chirped-poling nonlinear crystal, which is featured with a much larger phase-matching window[45,48]. Accordingly, the pump power needs to be augmented to maintain a high conversion efficiency in the case of using a larger beam size.

In our work, the single-pixel imaging is demonstrated with binary sparse objects. In principle, those related computational imaging methods could work well for non-sparse and non-binary objects in real scenarios. Moreover, a shorter acquisition time would be available for a smaller number of sampling measurement by using more advanced imaging processing algorithms. For instance, it has been shown that a partially sampled Hadamard matrix along with a regularized image reconstruction matrix can significantly reduce the computational time[36]. In addition, the presented imaging modality is equipped with a picosecond coincidence optical gating. The time-stamp functionality allows us to realize a high-resolution single-pixel depth imaging by temporally selecting the reflected photons[46,48]. With a combination of a tunable MIR source, our imaging system could further be extended to implement compressive four-dimensional spectro-volumetric imaging[52]. Thanks to recent advances in nonlinear materials and conversion techniques, the presented paradigm would provide a novel solution to the low-light imaging in the long-infrared[53] and terahertz[54] regions, where focal plane detector arrays are unavailable or prohibitively expensive. We believe that the presented approach of MIR single-photon computational imaging would promote applications including photon-starving infrared sensing, opaque material imaging, and phototoxicity-free biomedical examination.

## Methods

### Dual-color laser sources

All the light sources in the experiment originate from a passively synchronized fiber laser system that consists of two Er- and Yb-doped fiber lasers (EDFL and YDFL). The two fiber lasers are mode-locked at a repetition rate of 14.6 MHz. One portion of the YDFL output is amplified to obtain a higher average power of 280 mW, and serves as the pump source. The amplified spectrum is centered at 1030 nm with a bandwidth of 0.3 nm. The other branch of YDFL is mixed with the EDFL at 1550 nm to perform the difference frequency generation, which resulting in a MIR source at 3070 nm with a bandwidth of 0.9 nm. The signal and pump sources are then sent into the single-pixel upconversion imaging system based on sum-frequency generation. The narrow-band laser sources are favorable to fulfill the phase-matching condition for a high nonlinear conversion efficiency. Additionally, the pulse duration of the pump is tailored to be longer than that of the signal. Consequently, the MIR photons are temporally confined within the central part of the pump intensity envelope, thus improving the total conversion efficiency.

### Operating configurations

The hardwares for our MIR single-pixel imaging system are mainly comprised of spatial light modulators, single-element detectors, and control/acquisition units. The spatial modulation is conducted via a DMD (Texas Instrument, DLP650LNIR), which consists of $1280 \times 800$ micromirrors with a pixel pitch of $10.8$-$\mu$m and on/off tilting angles of $\pm 12°$. A high-speed frame switching is allowed to operate at a binary pattern rate up to 10.752 kHz. The DMD is specified to operate at the near-infrared band from 800 to 2000 nm for ensuring an efficient beam steering performance. Two types of detectors are used in our experiment. One is an analog silicon optical detector (Thorlabs, PDA100A2) with a minimum noise equivalent power about 3 pW/Hz$^{1/2}$, a response bandwidth up to 11 MHz, and an active area of 75.4 mm$^2$. The other is a digital Si-based photon counter (Excelitas, SPCM-AQRH-54), as specified with a 63% detection efficiency at 771 nm, a dark noise below 100 Hz, and a 180-$\mu$m sensor diameter. The detector outputs are registered by a data acquisition system based on a FPGA (Altera, Cyclone II) with a sampling clock up to 250 MHz and a 12-bit analog-to-digital converter (Analog Devices, AD7091R8) with a sampling time of 1 $\mu$s. The timing sequences for synchronization trigger, signal acquisition, and data saving are controlled by the FPGA unit.

### Reconstruction algorithms

For the Hadamard encoding, the intensity-based imaging masks are prepared by subtraction between two sequential complementary patterns. The differential intensity measurement is beneficial to eliminate the unwanted low-frequency noises. Thanks to the orthonormal property of the Hadamard matrix, the inverse matrix is simply obtained via the transpose operation, which facilitates a fast image reconstruction. In the compressive imaging, the random patterns are used due to the incoherence to the spatial features of the target, which is essential for the sub-Nyquist sampling. The recovery of sparse signals can be realized based on compressed sensing algorithms, which usually minimize the $l_1$ norm in the convex optimization. To reduce Gaussian noise in the reconstructed image, a machine learning approach is used to implement a deep denoiser based on convolutional neural networks[49]. Note that more advanced imaging processing algorithms are available to further improve the restoration performance with a reduced undersampling ratio. Inevitably, more computational resources are required, which may result in a longer reconstruction time than that for the data acquisition.

## Data availability

The data that support the findings of this study are available from the corresponding author upon request.

## Code availability

The codes for image reconstruction are available from the corresponding author upon request.

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

## Acknowledgements

This work was supported by the National Key Research and Development Program (2021YFB2801100), National Natural Science Foundation of China (62175064, 11621404, 11727812), Shanghai Municipal Science and Technology Major Project (2019SHZDZX01), and Fundamental Research Funds for the Central Universities.

## Author contributions

K.H. and H.Z. conceived the idea and designed the experiments. Y.W., J.F., and K.H. built the system, performed experiments, and processed data. M.Y. built fiber laser sources. E W. analyzed the imaging data. Y.W.

and K.H. wrote the manuscript draft. All authors were involved in discussions and contributed to the manuscript editing.

## Competing interests

The authors declare no competing interests.
