## [Peer Review File · Nature Communications]

Mid-infrared single-pixel imaging at the single-photon levelREVIEWER COMMENTS

Reviewer #1 (Remarks to the Author):

The work is oriented towards imaging in the MIR with a silicon detector at the single-photon level and at room temperature.

Based on nonlinear sum-frequency mixing (SFM) upconversion, the authors extend single-pixel imaging to the mid infrared (MIR) by shifting the pixel signal to a spectral region where there are well performing silicon detectors, including avalanche photodiodes (APDs) that can detect light down to the single-photon level.

Upconversion detection is a well-known technique to assist and improve detection in the infrared both in 1D with simple photodiodes, and in 2D with focal plane array (FPA) cameras.

In general, single pixel imaging is based on 1D detection. It relies on a transversal modulation or convolution of the original image using a set of orthonormal patterns that form a basis for expansion of the image (the Hadamard patterns for instance), with each pattern providing a scalar value in the detection pixel that determines the weight of a given pattern for the reconstruction of the image. In general, decomposition of the image in single-pixel imaging is accomplished by generating the test patterns with a spatial light modulator (SLM) or a Digital Micromirror Device (DMD).

The main novelty in this work is in my opinion that instead of applying the image decomposition patterns directly to the original infrared image, the patterns are alternatively imposed on the transversal profile of the pump beam required for SFM. Because SLMs and DMDs can not operate with good performance in the MIR, imposing the patterns in the pump beam facilitates the technique. With the technique reported by the authors, the use of a SLM or a DMD is not avoided. However, these devices need only to operate in the near infrared around 1030 nm where its operational characteristics are better than in the MIR. Based on the use of an APD and photon counting, the authors also report sparse imaging in the MIR.

The paper is clearly written, well presented, of high scientific quality, and well referenced. There are interesting contributions from the authors in this work, as expected in scientific/technological journal papers, and no question their work deserves publication in a journal. The authors show a deep understanding of few photon detection. However, in my opinion this work is not of an impact high enough to represent a scientific or technological breakthrough with the expected standards of Nat. Comm. Here is the reason:

Early in the manuscript, the authors state that "To date, it remains a long-standing quest to achieve MIR single-photon imaging at room temperature". This statement does not seem accurate in view of the recent publication by some of the co-authors of this work Ref [47]. The results in Ref [47] seem not only to invalidate their statement, but also to surpass those presented in this work, and very presumably those that could result by comparing the ultimate potential of both techniques in the single-photon regime. A major breakthrough like single-photon imaging at room temperature in the MIR was already reported in Ref [47], K. Huang et al., "Wide-field mid-infrared single-photon upconversion imaging", Nat. Commun. 13, 1077 (2022)". Related to this breakthrough there is also Ref [46] by J.S. Dam et al., "Room-temperature mid-infrared single-photon spectral imaging", Nat. Photon. 6, 788-793 (2012). In my opinion, as the breakthrough of achieving MIR single-photon imaging at room temperature based on image upconversion has been achieved previously, the authors need to support or demonstrate clearly that the potential performance of their alternative technique oriented to MIR single-photon imaging at room temperature can clearly surpass that of Ref [47]. This means either in terms of final resolution, conversion efficiency, or image-forming speed.

The authors also state that the computational upconversion imaging modality in this work offers practical and economic advantages, but this point does not seem clear either. As they mention, they exclude the need for slow mechanical scanning or expensive multipixel detectors (I interpret that

multipixel detectors here refers to MIR cooled multipixel detectors). The capability of achieving single-photon detection in upconversion with an APD is also shared by EMCCDs in 2D upconversion imaging. In fact, in ref [46], an EMCCD is used to detect an upconverted image in the MIR at the single-photon level and at room temperature. The economic advantage of an APD plus a SLM or a DMD versus an EMCCD does not seem clear. An APD has a lower cost than an EMCCD camera for sure, but the SLM or a DMD required for the single-pixel technique may have a cost comparable to that of an EMCCD. Thus, in my opinion, price and system complexity are not clearly an advantage with the technique here reported, and so far the results presented regarding resolution, image forming speed and conversion efficiency are poorer. From the point of view of simplicity, the set-up described does not seem simpler than that of Ref [47].

Regarding the performance in resolution, only a 64x64 pixel image is obtained. In page 7, the authors indicate that this resolution can be increased:

“We note that the achieved spatial resolution is technically limited by the numerical aperture of the upconversion imaging system, where the acceptance angle for the incident light is truncated by the phase-matching angular bandwidth of the nonlinear crystal [39]. Additionally, the field of view is determined by the pump diameter and/or crystal aperture size, whichever is smaller. Hence, the use of a shorter crystal with a wider section would lead to a better spatial resolution and a larger field of view. An alternative approach to enhance the spatial resolution can resort to a chirped-poling nonlinear crystal, which is featured with a much larger phase-matching window [44, 47]. Accordingly, the pump power needs to be augmented to maintain a high conversion efficiency in the case of using a larger beam size”.

The limitation in resolution seems serious due to the QPM acceptance angle of the set-up (mainly due to periodical poling) according to the authors. However, some of the coauthors have recently overcome this limitation by using a chirped PPLN crystal in Ref [47]. This makes one wonder why the authors have not used then the same chirped crystal used in Ref [47] and have presented better resolution results in this work.

Presumably, the ultimate potential performance comparison between this work and that in Ref [47] would involve considering crystal aperture (equal in both cases), focusing, SLM or DMD pixel size and speed limitations at the pump wavelength, most adequate domain distribution pattern in each case....

In summary, my main concern for considering an impact high enough to publish this work in Nat. Comm. rather than in another journal is whether it can be supported theoretically, experimentally, or plausibly argued, that the new technique reported in this work can perform better than that used in Ref [47]. It would also be convenient to justify well that the single-pixel technique with APD photon counting plus a SLM or DMD, leads to a significant lower cost or complexity in implementing as compared to using an EMCCD.

A closely related reference to the part of this work devoted to “Single-photon compressive imaging” regarding the use of pulse coincidence for enhancing conversion efficiency and allowing time-of-flight imaging, although using different wavelengths relies in K. Huang et al., “Few-photon-level two-dimensional infrared imaging by coincidence frequency upconversion”, Appl. Phys. Lett. 100, 151102 (2012). This reference not included in the manuscript could perhaps be of interest for some potential readers.

Reviewer #2 (Remarks to the Author):

Review.

The paper, Mid-infrared single-pixel imaging at the single-photon level, combines for the first time, “single-pixel imaging” and “upconversion”, to demonstrate SWIR (@ 3 μm range), single-photon imaging. Typical image size is 16x16 pixel at 10 Hz update rate. A highlight is the impressively low noise which allows image formation even with illumination at the 1 photon/pulse level.

The paper is clearly and well drafted, with easy explanations and high quality graphs/figures. The same is the case for supplementary material. The paper is technically interesting and inspirational to read.

Single-pixel imaging as a concept is intriguing and likely of interest for a broader readership. Typical arguments for using single-pixel imaging is to provide images at wavelengths where detector arrays such as CCDs and mid-IR focal plane arrays are not available or have poor performance (besides more technical aspects). Zooming in on the SWIR region, focal plane arrays are available with many more pixels than 16x16 and higher update rate. Therefore the technical edge of the proposed method should then be found in its single photon capability (mainly), the possibility to time gate signals, as well as room temperature operation of the system.

Novelty.

The combination of single-pixel imaging and upconversion is novel. In the literature, upconversion is often/mostly included for translating mid-IR light to the near infrared region for easy CCD array detection/imaging. Therefore it appears counter-intuitive to combine “single-pixel imaging” with “upconversion” since both technologies address the same problem(!). E.g. [46] demonstrates single photon imaging in the 3 μm range using upconversion, however using a high-end and expensive iXon camera. In contrast to this work, [46] can use incoherent SWIR illumination and works well in the CW regime. The CW operation of [46] did not have the time gating functionality available in contrast to this work, therefore the single-photon sensitivity demonstrated here is by far better - which is a highlight. Comparing the two different approaches is difficult, however the technical complexity of [46] is smaller, e.g. no need for ps, synchronised pump and tunable IR lasers, DMD, as well as complex post processing. In summary and as mentioned, I primarily find this systems single-photon imaging performance unique and impressive.

- I suggest that the authors include a sentence emphasising the fast response time of upconversion, which supports the time gating used to ensure noise suppression.

Spatial resolution.

The spatial resolution is not “impressive”. The authors do not mention an actual spatial resolution in the form of number of resolvable spatial elements, however noting that the acceptance angle of the ppLN crystal as being a limiting factor. I do agree with that.

- It could be interesting to understand the influence of the ppLN crystal acceptance angle better (including the actual acceptance angle in the submission) so as to understand what parameters determines the resolution of the proposed method.

To open that discussion, I would expect that the high spatial angles emerging from the super imposed spatial structure(s) on the 1.03 μm pump source (including degradation from the MEMS modulator) to be at least one main concern WHEN combined with the limited acceptance angle of the crystal. I would expect that using a shorter ppLN crystal, thus larger acceptance angle, will lead to better spatial resolution. That said, this will come at the expense of efficiency, but maybe the low noise performance of the system can make up for that (?).

Efficiency.

I cannot find an evaluation of the systems efficiency including the QE of the upconversion process. Efficiency is only illustrated at systems level in the form of demonstrated single photon imaging, strongly improved by timing of photons. My impression will be that “high power” pulses are needed in the presented system.

- Further discussion of QE could be included

Conclusion.

I find the submitted paper very interesting and worth reading. The technical complexity of the presented system (best presented in the supplementary material) likely makes this system mostly relevant for special, high-end, IR applications and research. Including further information on the three bullet points will improve the manuscript.

Manuscript NCOMMS-22-36840
“Mid-infrared single-pixel imaging at the single-photon level”
Reply to the Reviewers

We would like to thank the two reviewers for the careful reading of the manuscript and their valuable reports. We give below a detailed response to the reviewers’ comments. Excerpts from the original reports are given in blue. Changes in the revised manuscript are indicated in green.

Reviewer #1

The work is oriented towards imaging in the MIR with a silicon detector at the single-photon level and at room temperature.

Based on nonlinear sum-frequency mixing (SFM) upconversion, the authors extend single-pixel imaging to the mid infrared (MIR) by shifting the pixel signal to a spectral region where there are well performing silicon detectors, including avalanche photodiodes (APDs) that can detect light down to the single-photon level.

Upconversion detection is a well-known technique to assist and improve detection in the infrared both in 1D with simple photodiodes, and in 2D with focal plane array (FPA) cameras.

In general, single pixel imaging is based on 1D detection. It relies on a transversal modulation or convolution of the original image using a set of orthonormal patterns that form a basis for expansion of the image (the Hadamard patterns for instance), with each pattern providing a scalar value in the detection pixel that determines the weight of a given pattern for the reconstruction of the image. In general, decomposition of the image in single-pixel imaging is accomplished by generating the test patterns with a spatial light modulator (SLM) or a Digital Micromirror Device (DMD).

The main novelty in this work is in my opinion that instead of applying the image decomposition patterns directly to the original infrared image, the patterns are alternatively imposed on the transversal profile of the pump beam required for SFM. Because SLMs and DMDs cannot operate with good performance in the MIR, imposing the patterns in the pump beam facilitates the technique. With the technique reported by the authors, the use of a SLM or a DMD is not avoided. However, these devices need only to operate in the near infrared around 1030 nm where its operational characteristics are better than in the MIR. Based on the use of an APD and photon counting, the authors also report sparse imaging in the MIR.

We thank the reviewer for his/her careful reading of the manuscript, and for the recognition of the novelty of our work. As the reviewer pointed out, the upconversion imaging is typically realized by using focal plane array cameras. In contrast, our work reports on a novel implementation of mid-infrared single-photon imaging based on a single-element detector. The underlying dual functionalities of the nonlinear pump modulation not only facilitates the high-fidelity mapping of time-varying patterns onto the infrared radiation, but also enables the sensitive upconversion detection with a silicon

bucket detector operating at room temperature. It is the brand-new concept of nonlinear structured detection that enables us to address the challenge toward single-photon computational imaging in the mid-infrared regime. The presented paradigm of single-pixel upconversion imaging establishes a new path for sensitive imaging at longer infrared wavelengths or terahertz frequencies, where the high-sensitivity photon counter and high-fidelity spatial modulator are typically hard to access.

The paper is clearly written, well presented, of high scientific quality, and well referenced. There are interesting contributions from the authors in this work, as expected in scientific/technological journal papers, and no question their work deserves publication in a journal. The authors show a deep understanding of few photon detection. However, in my opinion this work is not of an impact high enough to represent a scientific or technological breakthrough with the expected standards of Nat. Comm. Here is the reason:

We thank the reviewer for the positive evaluation on our manuscript. The achievements in our manuscript are considered to be well supported by the solid data and clear plots. As pointed out by the reviewer, our work contributes to deep understanding of the few-photon detection. Indeed, the ingenious connection between the upconversion technique and structured pump modulation opens a new path to realizing the single-photon mid-infrared imaging with a single-element silicon-based detector. This novel modality based on the nonlinear structured detection allows us to demonstrate for the first time the mid-infrared single-pixel imaging at the single-photon level.

It is worth noting that there are no other setups enabling this performance heretofore. By itself, this state-of-the-art performance is therefore a landmark advance in the field of single-pixel imaging. In contrast to previous MIR upconversion imaging configurations, using a single-pixel detector instead of a pixelated array sensor favors to realize unique detection features with a fast response time and a single-photon sensitivity. This is a remarkable advance and an enabling tool for many promising applications, particularly related to those requiring the mid-infrared imaging with a precise timing and a high sensitivity. We believe that our work is significant and of broad interest and, as such, it fits well the scope of Nature Communications.

In the following, we give answers to the points raised by the reviewer and describe the changes we made in the manuscript to take into account these valuable comments, which helped us to further clarify the novelty and improve the clarity of the paper.

Early in the manuscript, the authors state that “To date, it remains a long-standing quest to achieve MIR single-photon imaging at room temperature”. This statement does not seem accurate in view of the recent publication by some of the co-authors of this work Ref [47]. The results in Ref [47] seem not only to invalid their statement, but also to surpass those presented in this work, and very presumably those that could result by comparing the ultimate potential of both techniques in the single-photon regime. A major breakthrough like single-photon imaging at room temperature in the MIR was already reported in Ref [47], K. Huang et al., “Wide-field mid-infrared single-photon upconversion imaging”, Nat. Commun. 13, 1077 (2022)”. Related to this breakthrough there is also Ref [46] by J.S. Dam et al., “Room-temperature mid-infrared

single-photon spectral imaging”, Nat. Photon. 6, 788-793 (2012). In my opinion, as the breakthrough of achieving MIR single-photon imaging at room temperature based on image upconversion has been achieved previously, the authors need to support or demonstrate clearly that the potential performance of their alternative technique oriented to MIR single-photon imaging at room temperature can clearly surpass that of Ref [47]. This means either in terms of final resolution, conversion efficiency, or image-forming speed.

The statement is placed at the end of the first paragraph in the introductory part, which attempts to make a general remark on the status quo of direct mid-infrared imagers based on narrow bandgap semiconductors. Nowadays, continuous endeavors have still been dedicated to approaching sensitive direct detection for MIR photons especially at room temperature, yet the reported state-of-the-art imaging sensitivity for HgCdTe or InSb cameras is about nW, which is many orders of magnitude far away from the single-photon level. Hence, it remains a long-standing quest to achieve direct MIR imaging at the single-photon level under the room-temperature condition.

We agree with reviewer that frequency upconversion technique provides a feasible way to realize the single-photon mid-infrared imaging. The indirect scheme for sensitive mid-infrared detection has indeed witnessed great processes in recent years, as exemplified by the two mentioned works. In all previous demonstrations including the two referenced works, a high-end multipixel camera is typically required for the infrared upconversion imaging. In contrast, our work proposed and implemented a novel upconversion imaging modality based on nonlinear structured detection, which leads to the first demonstration of mid-infrared single-pixel imaging at the single-photon level. The scope of this work concentrates in advancing the frontier of single-pixel imaging paradigm into the mid-infrared regime, which is conceptually different from previously upconversion imagers based on pixelated devices.

In addition to the conceptual novelty, the presented mid-infrared imager naturally inherits desirable strengths of the single-pixel camera. In the discussion section of the manuscript, we have already presented a direct comparison for the two strategies in the context of upconversion imaging techniques, which highlights the unique features of the presented mid-infrared single-pixel imager. Specifically, the operation bandwidth for single-element detectors and corresponding timing electronics are vastly superior to that for current pixelated devices. The better timing resolution favors to implement time-of-flight ranging or three-dimensional imaging, where the information of precise arrival time for detected photons is required.

We thank the reviewer’s valuable comments, and have revised the manuscript for better clarity and strengthened novelty. In the introductory part, the statement now wrote as: *“To date, it remains a long-standing quest to achieve single-photon direct MIR imaging at room temperature.”* The scope of our work as well as the advantageous features are elaborated: *“So far, MIR single-photon computational imaging has yet been realized to reveal the full potential of the single-pixel paradigm, which thus urgently calls for the development of techniques to address the challenges on the single-photon detection and high-resolution modulation at MIR wavelengths.”*, *“..., the operation bandwidth for single-element detectors and corresponding timing electronics are vastly superior to that for current pixelated devices \cite{Edgar2019NP}. The better timing resolution favors to implement time-of-flight ranging or three-dimensional imaging, where the information of precise arrival time for detected photons is required \cite{Kirmani2014Science}.”*

The authors also state that the computational upconversion imaging modality in this work offers practical and economic advantages, but this point does not seem clear either. As they mention, they exclude the need for slow mechanical scanning or expensive multipixel detectors (I interpret that multipixel detectors here refers to MIR cooled multipixel detectors). The capability of achieving single-photon detection in upconversion with an APD is also shared by EMCCDs in 2D upconversion imaging. In fact, in ref [46], an EMCCD is used to detect an upconverted image in the MIR at the single-photon level and at room temperature. The economic advantage of an APD plus a SLM or a DMD versus an EMCCD does not seem clear. An APD has a lower cost than an EMCCD camera for sure, but the SLM or a DMD required for the single-pixel technique may have a cost comparable to that of an EMCCD. Thus, in my opinion, price and system complexity are not clearly an advantage with the technique here reported, and so far the results presented regarding resolution, image forming speed and conversion efficiency are poorer. From the point of view of simplicity, the set-up described does not seem simpler than that of Ref [47].

Indeed, the presented imaging system uses a single-element detector and a spatial light modulator, as typically required for all single-pixel cameras. In general, the price of a SLM or a DMD is much cheaper than the EMCCD. For instance, the price of the DMD (Texas Instrument, DLP650LNIR) used in our experiment is less than \$10k, and the price for the APD (Excelitas, SPCM-AQRH-54) is about \$6k, while the price of the EMCCD (Andor, iXon Ultra 888) used in Ref. [47] reaches to over \$80k. It is the relative economic budget along with other appealing features that renders the single-pixel architecture as an attractive imaging modality.

Regarding to the “single-pixel simplicity” stated in our manuscript, it refers to the use of a silicon avalanche photodiode operating at room temperature, which is apparently much simpler than MIR cooled multipixel detectors in terms of device fabrication and operation. In our proof-of-principle demonstration, the experiment setup is designated to conveniently play with system parameters for better investigating the single-pixel performance, which increases the complexity in lasers and optics. These experimental settings enable us to well demonstrate the novel concept of nonlinear structured detection, and to facilitate for the first time the mid-infrared single-pixel imaging at the single-photon level. The proposed imaging modality would stimulate a broad interest, and inspire other variants for further optimizing the imaging performance.

Regarding the performance in resolution, only a 64x64 pixel image is obtained. In page 7, the authors indicate that this resolution can be increased:

“We note that the achieved spatial resolution is technically limited by the numerical aperture of the upconversion imaging system, where the acceptance angle for the incident light is truncated by the phase-matching angular bandwidth of the nonlinear crystal [39]. Additionally, the field of view is determined by the pump diameter and/or crystal aperture size, whichever is smaller. Hence, the use of a shorter crystal with a wider section would lead to a better spatial resolution and a larger field of view. An alternative approach to enhance the spatial resolution can resort to a chirped-poling nonlinear crystal, which is featured with a much larger phase-matching window

[44, 47]. Accordingly, the pump power needs to be augmented to maintain a high conversion efficiency in the case of using a larger beam size”.

The limitation in resolution seems serious due to the QPM acceptance angle of the set-up (mainly due to periodical poling) according to the authors. However, some of the coauthors have recently overcome this limitation by using a chirped PPLN crystal in Ref [47]. This makes one wonder why the authors have not used then the same chirped crystal used in Ref [47] and have presented better resolution results in this work.

Presumably, the ultimate potential performance comparison between this work and that in Ref [47] would involve considering crystal aperture (equal in both cases), focusing, SLM or DMD pixel size and speed limitations at the pump wavelength, most adequate domain distribution pattern in each case....

As the reviewer pointed out, the spatial resolution in our imaging system is limited by the acceptance angle of the nonlinear crystal. Generally, there is always a trade-off between the phase-matching bandwidth and the nonlinear conversion efficiency. As such, the conversion efficiency for a chirped-poling nonlinear crystal is typically two orders of magnitude lower than that for a uniform-grating crystal at the condition of an identical pump intensity. Hence, a periodically-poled lithium niobate (PPLN) crystal is used to facilitate a higher conversion efficiency in our experiment. Limited by the currently available average power from the laser system, the pump power of 280 mW is used to illuminate the DMD to prepare the structured optical patterns. In this power setting, the quantum conversion efficiency is measured to be about 1% for an “all-white” pump pattern. The modest efficiency is sufficient for us to demonstrate the single-photon imaging performance owing to the extremely low background noise.

We note that the pulse durations for the lasers used in our experiment is about 30 ps, which is much longer than those in Ref. [47]. The reduced peak intensity of the pump leads to a smaller conversion efficiency. For the sake of demonstrating the photon-sparse imaging at the mid-infrared, the PPLN crystal is thus used in this work. Further improvement on the conversion efficiency is possible by boosting the pump power. Notably, a high-efficiency spatial modulator is required to reduce the risk of laser damage due to the intensive illumination. Under sufficiently intensive pump, the chirped-poling crystal can be used to significantly enhance the spatial resolution and the field of view. This improvement will be our future work toward more advanced imaging performance.

Note that the scope of our work is to provide a novel imaging modality over the previously upconversion configurations based on multipixel sensors, and to establish an effective solution for implementing the mid-infrared computational imaging at the single-photon level. The mid-infrared single-pixel camera would be particularly useful in subsequent applications that require high detection sensitivity and fast time response.

In Supplementary Note 2 of the revised version, we added more details on the nonlinear conversion performance, and made a comparison to our previous work: *“In the experiment, the pump power impinging onto the digital micromirror device (DMD) is set to be about 280 mW, which is below the damage threshold of the spatial modulator. The corresponding conversion efficiency is measured to be about 1% by calculating the photon-flux ratio between the input infrared light and the upconverted*

SFG light. The modest efficiency is sufficient for us to demonstrate the single-photon imaging performance thanks to the extremely low background noise. Further improvement on the conversion efficiency is possible by boosting the pump power in combination with a high-efficiency spatial modulator. Here, the pulse duration for the pump laser is about 30 ps, which is much longer than that in our previous work \cite{Huang2022NC}. The reduced peak intensity of the pump pulse leads to the decrease of the conversion efficiency. Further improvement on the conversion efficiency can resort to the use of femtosecond pump pulses.”

In summary, my main concern for considering an impact high enough to publish this work in Nat. Comm. rather than in another journal is whether it can be supported theoretically, experimentally, or plausibly argued, that the new technique reported in this work can perform better than that used in Ref [47]. It would also be convenient to justify well that the single-pixel technique with APD photon counting plus a SLM or DMD, leads to a significant lower cost or complexity in implementing as compared to using an EMCCD.

As stated above, the mentioned work in Ref. [47] focuses on the improving the field of view for the upconversion imaging system based on a multipixel camera, which is conceptually different from our scope here on establishing a new path to realizing the ultrasensitive single-pixel imaging at the mid-infrared. It is worth noting that although the single-pixel architecture has widely been demonstrated to perform imaging over a broad electromagnetic spectrum, yet single-pixel cameras with single-photon sensitivity have long been restricted to operate at the visible or near-infrared wavelengths due the availability of high-sensitivity photon detectors and high-fidelity spatial modulators. To date, it remains a challenge to achieve single-photon computational imaging in the mid-infrared regime.

In this work, we have addressed the long-standing quest by proposing a so-called nonlinear structured detection approach, and demonstrated for the first time the mid-infrared single-pixel imaging at the single-photon level. The achieved imaging sensitivity is beyond the performance for any reported mid-infrared single-pixel cameras heretofore, which thus represents a significant and firm advance in the single-element imaging architecture. The distinct feature for single-pixel detector is the much faster time response comparing to the pixelated imaging device. Therefore, the implemented single-pixel imager is particularly useful in time-resolved imaging. In view of the novel concept, original implementation, and advanced results, we believe that the manuscript fits well the standards of Nature Communications.

We thank again the referee for having given us the opportunity to clarify few points of our paper and to make more explicitly the novelty of our work. We hope with these changes he/she might consider our paper for acceptance.

A closely related reference to the part of this work devoted to “Single-photon compressive imaging” regarding the use of pulse coincidence for enhancing conversion efficiency and allowing time-of-flight imaging, although using different wavelengths relies in K. Huang et al., “Few-photon-level two-dimensional infrared imaging by coincidence frequency upconversion”, Appl. Phys. Lett. 100, 151102 (2012). This reference not included in the manuscript could

perhaps be of interest for some potential readers.

We thank the reviewer to point out this work, and have added the reference in the revised manuscript.

Reviewer #2

Review.

The paper, Mid-infrared single-pixel imaging at the single-photon level, combines for the first time, “single-pixel imaging” and “upconversion”, to demonstrate SWIR (@ 3 μm range), single-photon imaging. Typical image size is 16x16 pixel at 10 Hz update rate. A highlight is the impressively low noise which allows image formation even with illumination at the 1 photon/pulse level.

The paper is clearly and well drafted, with easy explanations and high quality graphs/figures. The same is the case for supplementary material. The paper is technically interesting and inspirational to read.

Single-pixel imaging as a concept is intriguing and likely of interest for a broader readership. Typical arguments for using single-pixel imaging is to provide images at wavelengths where detector arrays such as CCDs and mid-IR focal plane arrays are not available or have poor performance (besides more technical aspects). Zooming in on the SWIR region, focal plane arrays are available with many more pixels than 16x16 and higher update rate. Therefore the technical edge of the proposed method should then be found in it’s single photon capability (mainly), the possibility to time gate signals, as well as room temperature operation of the system.

We thank the reviewer for his/her positive evaluation and helpful comments on our work. In the following, we present the reply and related changes in the revised manuscript.

Novelty.

The combination of single-pixel imaging and upconversion is novel. In the literature, upconversion is often/mostly included for translating mid-IR light to the near infrared region for easy CCD array detection/imaging. Therefore it appears counter-intuitive to combine “single-pixel imaging” with “upconversion” since both technologies address the same problem(!). E.g. [46] demonstrates single photon imaging in the 3 μm range using upconversion, however using a high-end and expensive iXon camera. In contrast to this work, [46] can use incoherent SWIR illumination and works well in the CW regime. The CW operation of [46] did not have the time gating functionality available in contrast to this work, therefore the single-photon sensitivity demonstrated here is by far better - which is a highlight. Comparing the two different approaches is difficult, however the technical complexity of [46] is smaller, e.g. no need for ps, synchronised pump and tunable IR lasers, DMD, as well as complex post processing. In summary and as mentioned, I primarily find this systems single-photon imaging performance unique and impressive.

- I suggest that the authors include a sentence emphasising the fast response time of upconversion, which supports the time gating used to ensure noise suppression.

We thank the reviewer’s insightful suggestion. In the revised manuscript, we add the related discussion

on the fast response feature: *“Moreover, the instantaneous response of the nonlinear upconversion enables us to apply the synchronous ultrashort optical gating for the signal detection, thus significantly reducing the background noise from the ambient random scattering.”*

Spatial resolution.

The spatial resolution is not “impressive”. The authors do not mention an actual spatial resolution in the form of number of resolvable spatial elements, however noting that the acceptance angle of the ppLN crystal as being a limiting factor. I do agree with that.

- It could be interesting to understand the influence of the ppLN crystal acceptance angle better (including the actual acceptance angle in the submission) so as to understand what parameters determines the resolution of the proposed method.

To open that discussion, I would expect that the high spatial angles emerging from the super imposed spatial structure(s) on the 1.03 μm pump source (including degradation from the MEMS modulator) to be at least one main concern WHEN combined with the limited acceptance angle of the crystal. I would expect that using a shorter ppLN crystal, thus larger acceptance angle, will lead to better spatial resolution. That said, this will come at the expense of efficiency, but maybe the low noise performance of the system can make up for that (?).

We agree with the reviewers that the achieved spatial resolution is indeed limited by some technical imperfections in our imaging system. Figure R1 illustrates the upconversion imaging configuration in our experiment, where the nonlinear spectral conversion takes place in an intermediate image plane at the presence of a structured pump. It can be seen that the numerical aperture of the imaging system is determined by the phase-matching angular bandwidth of the nonlinear crystal. Specifically, the minimal resolved size d_{res} in the image plane is given by [R1]

$$d_{res} \approx \frac{2 \ln(2) \lambda_{IR}}{\pi \psi_{IR}} ,$$

where ψ_{IR} corresponds to the acceptance angle, λ_{IR} is the wavelength for the incident infrared field. Meanwhile, the field of view of the imaging system is restricted to the beam size of the pump.

Figure R1: Scheme of the single-pixel upconversion imaging system in our experiment.

In our experiment, the transverse size of the nonlinear crystal is 1 mm. The pump diameter is adapted

to be about 691 μm to avoid truncation by the crystal aperture, which resulted in about 8×8 resolvable spatial elements by taking into account of the measured spatial resolution of 91 μm . Accordingly, the acceptance angle is inferred to be about several degrees, which is limited by the stringent quasi-phase matching condition.

As the reviewer pointed out, the limited spatial resolution in the experiment is also subjected to the imperfect mapping of the structured pump mode from the spatial light modulator. Rigorously, the propagation evolution of the spatial distribution can be investigated by using the Collins diffraction integral equation as presented in Supplementary Note 1. In addition, some high spatial frequencies of the pump pattern are not involved in the efficient conversion process, will would further degrade the spatial resolution.

To go beyond the achieved space-bandwidth product, it is imperative to increase the phase-matching angular bandwidth of the nonlinear crystal and enlarge the beam size of confined pump within the crystal. Hence, the use of a shorter crystal with a wider section could lead to a better spatial resolution and a larger field of view. An alternative approach to enhance the spatial resolution can resort to a chirped-poling nonlinear crystal, which is featured with a much larger phase-matching window. The associated penalty of these two approaches is the reduced conversion efficiency. Thanks to the ultralow background noise due to the ultrashort optical gating, a faithful reconstruction in the photon-starving scenario would still be possible with a longer acquisition time.

[R1] A. Barh, P. J. Rodrigo, L. Meng, C. Pedersen, and P. Tidemand-Lichtenberg, "Parametric upconversion imaging and its applications," *Adv. Opt. Photon.* 11, 952 (2019).

In the revised manuscript, we have added relevant discussions for the limitation of the spatial resolution, and possible solutions to improve it: *“The exhibiting blurring effect is due to the reduced numerical aperture determined by the phase-matching bandwidth of the angular acceptance for the nonlinear crystal. Additionally, the limited spatial frequency bandwidth of the upconversion imaging system inevitably results in an imperfect mapping of the optical pump masks, which may contribute to the image degradation in the single-pixel imaging paradigm.” “Hence, the use of a shorter crystal with a wider section would lead to a better spatial resolution and a larger field of view. An alternative approach to enhance the spatial resolution can resort to a chirped-poling nonlinear crystal, which is featured with a much larger phase-matching window [45,48]. Accordingly, the pump power needs to be augmented to maintain a high conversion efficiency in the case of using a larger beam size.”*

Efficiency.

I cannot find an evaluation of the systems efficiency including the QE of the upconversion process. Efficiency is only illustrated at systems level in the form of demonstrated single photon imaging, strongly improved by timing of photons. My impression will be that “high power” pulses are needed in the presented system.

- Further discussion of QE could be included

We follow the reviewer's suggestion and add the discussion on the quantum efficiency. In the revised version, we have elaborated the relevant details in Supplementary Note 2: *"In the experiment, the pump power impinging onto the digital micromirror device (DMD) is set to be about 280 mW, which is below the damage threshold of the spatial modulator. The corresponding conversion efficiency is measured to be about 1% by calculating the photon-flux ratio between the input infrared light and the upconverted SFG light. The modest efficiency is sufficient for us to demonstrate the single-photon imaging performance thanks to the extremely low background noise. Further improvement on the conversion efficiency is possible by boosting the pump power in combination with a high-efficiency spatial modulator."*

Conclusion.

I find the submitted paper very interesting and worth reading. The technical complexity of the presented system (best presented in the supplementary material) likely makes this system mostly relevant for special, high-end, IR applications and research. Including further information on the three bullet points will improve the manuscript.

We thank again the reviewer for his/her positive report and valuable suggestions. In the revised manuscript, we have carefully addressed the raised points to improve our manuscript.

REVIEWERS' COMMENTS

Reviewer #2 (Remarks to the Author):

The authors have responded satisfactory to suggested specific changes/points in the review.